# Changing lifestyle for dementia risk reduction: Inductive content analysis of a national UK survey

**Alessandro Bosco** **[1]\*, Katy A. Jones[1], Claudio Di Lorito[2], Blossom C. M. Stephan[1], Martin Orrell[1], Deborah Oliveira[3]**

**1** Division of Psychiatry and Applied Psychology, University of Nottingham, Nottingham, United Kingdom,
**2** Division of Rehabilitation Ageing & Wellbeing, University of Nottingham, Nottingham, United Kingdom,
**3** Department of Psychiatry, Universidade Federal de Sao Paulo (UNIFESP), Sao Paulo, Brazil

\* alessandro.bosco@nottingham.ac.uk

## Abstract

**Data Availability Statement:** All relevant data are within the manuscript and its Supporting Information files.

### Objective

To explore what motivates individuals to change their behaviour to reduce their risk of dementia.

### Methods

We conducted secondary qualitative analysis of a UK-based online survey on motivation to change lifestyle and health behaviour for dementia risk reduction. Participants were recruited through social media, the Join Dementia Research network and the National Institute for Health Research Portfolio. Free-text comments from people aged ≥50 years were analysed by two researchers independently using inductive content analysis. Inter-rater agreement was measured through Cohen's Kappa coefficient.

### Results

Of the 3,948 participants completing the survey, 653 provided free text comments that were included in the analysis (Mean age = 64.1; SD = 8.3 years). The majority of the sample were women (n = 459; 70.3%), Caucasian (n = 625; 95.7%) and married/in partnership (n = 459; 70.3%). Three overarching themes were identified: (1) motivators to changing lifestyle; (2) barriers for lifestyle change; and, (3) quality of the information received. The inter-rater reliability of the coding was high ($k$ = 0.7). Having a family history of dementia or feeling like they had a healthy lifestyle already were motivating factors for behaviour change. Having competing health priorities other than dementia and caring for someone acted as de-motivators as they reduced the time available to dedicate to one's own health. Evidence-based information around dementia prevention was a motivator, but commonly the information was not trusted.

**Funding:** This is a secondary data analysis based on a dataset of a large scale survey funded by the Alzheimer's Research UK (Midland).

**Competing interests:** The authors have declared that no competing interests exist.

## Discussion

Aligned with the World Health Organisation (WHO) mandate on dementia prevention, community health campaigns targeting population awareness around behaviour change and dementia risk factor reduction are urgently needed. To be successful, such campaigns will need to be accompanied by individual approaches that can overcome age-related barriers and individual differences in motivation levels, personal barriers and trust in the information received.

## Introduction

Preventative strategies targeting modifiable risk factors to reduce the global burden of disease associated with dementia are an urgent public health priority [1]. This includes hypertension, physical inactivity, obesity, smoking, depression, low level of education, and diabetes [2, 3]. Reducing these risk factors by 10% could potentially half the number of people with dementia worldwide by 2050 [3]. However, research is needed to understand those factors which promote the lifestyle changes necessary to reduce dementia risk.

Worldwide, survey-based studies consistently report poor public awareness of the leading risk factors for dementia [4, 5, 6, 7, 8]. In the UK, a national survey of 2,361 adults aged ≥15 years, found that only 34% believed risk of dementia could be reduced through lifestyle changes, and 22% thought dementia was an inevitable consequence of ageing. In contrast, nearly three quarters were willing to receive more information from health professionals about personal risk factors for dementia [4].

Few studies have explored views about dementia prevention and engaging in health-promoting behaviours to reduce dementia risk [9]. In 'hard to reach' older people including the oldest old (people aged ≥85 years) and black and ethnic minorities, motivation to engage in healthy life choices (e.g. reducing alcohol intake and increasing physical activity levels) was linked to current activities, level of social engagement, and existing beliefs about health promotion [10]. Moreover, people living with multi-morbidity or in socially isolated areas were more reluctant to engage in health-promoting activities [10]. Hence, the way older individuals process and act on information about dementia prevention is complex and involves internal as well as external factors [11].

Public health campaigns around dementia prevention operate in a crowded media landscape. Evidence-based information needs to compete with unreliable and often sensationalist news that is provided in the form of catchy phrases, simple language, and through TV programmes that cater for large audiences, with little or no attention to the quality of the message being broadcast. A recent review by Cations et al. [9] found that dementia was commonly considered part of normal ageing despite public health efforts to debunk messages from untrusted media sources promoting the association. Kessler et al. [12] noted that, until individuals until individuals understand that there is a distinction between normal ageing and dementia, they will be reluctant to take personal responsibility and engage in any meaningful lifestyle changes to reduce dementia risk.

The WHO Global Action Plan [1] advocates for campaigns that could potentially buffer the noise brought by untrusted sources by, for example, designing both local and national population-specific (e.g. migrant groups) campaigns that make creative use of health promoting messages. However, before this is possible, there is a need to understand how to improve the efficacy and feasibility of lifestyle based public health programs for dementia risk reduction, and what may motivate individuals to improve their lifestyle. Given this, the aim of this

qualitative study is to explore willingness to change lifestyle and behavioural factors to reduce risk of future dementia in people aged ≥50 years.

## Methods

This was a secondary qualitative content analysis of an anonymous online survey of people's willingness to change lifestyle to reduce their risk of dementia. Full details of the survey methods are published (*reference blinded for peer review*). This study received ethical approval from *(reference and number blinded for peer review)*.

### Sample

In total, 3,948 participants aged ≥50 years (mean = 62.0, standard deviation (SD) = 8.0; range 50–93 years) were recruited nationally via social media, the Join Dementia Research platform (a UK-based online platform for participation in dementia research involving mostly people without dementia), and the National Institute for Health Research (NIHR) Portfolio. The majority were Caucasian (n = 3,805; 97.1%), lived in England (n = 3,586; 90.8%) and were women (n = 2,880; 72.9%). All individuals registering their interest to take part in the study were included unless they had dementia at the time of recruitment.

### Qualitative question (free-text comment)

Information on sociodemographic, current lifestyle and health status were collected. Using a five-point Likert scale, participants were also asked to rate their willingness to change their behaviour. For example, in the case of smokers, they were asked to rate to what extent they were willing to stop smoking to potentially reduce their risk of having dementia in the future.

At the end of the survey, participants were asked "Would you like to tell us anything about your experience in taking part in this survey?". Each participant had 800 characters to report on their experiences. They were not provided with any additional prompts and answering this question was not mandatory. Of the total sample, 1,158 (29.3%) provided a free-text response to this question and of these, 653 provided their views and experiences about changing behaviour for dementia risk reduction and were included in the data analysis reported in this paper. As the concept of health is socially constructed, we did not apply any preconceived definition for the term, and we used inverted commas to highlight the term in the results section to describe specific comments made by respondents on behaviour change and risk reduction.

### Data analysis

Data was analysed using iterative and inductive content analysis techniques [13]. Data analysis was conducted by (AB), with the support and advice from the co-authors at every stage.

In the preparation phase, the survey comments were exported to an Excel® file and read multiple times. Only narratives involving participants' views about dementia risk reduction and behaviour change were considered to be relevant for this paper (*n* = 653, 56.4%). Comments that were not related to the topic of interest (*n* = 368, 31.8%) were about the size of the comment box or about the survey questions (*n* = 152), the survey comprehensiveness (*n* = 111), preparedness to answer the survey questions (*n* = 38), perceived improvement in the knowledge of dementia prevention after taking part in the study (*n* = 32), and other comments (*n* = 35).

Following the preparation phase, open coding was conducted through the addition of notes next to each text box. Such notes were meant to represent condensed meaning units of the text. Those notes sharing similar content were then collated under higher order headings for

the development of codes (labels representing the meaning units). Newly formed codes were then exported to NVivo® 12 [14], so as to create higher order categories attained by clustering data according to the specific group they to which they belonged. In the final abstraction phase, a description for each developed category was generated and each category was named based on the information it contained. Subcategories were formed within each category when there was data that shared similar content that allowed for further categorisation.

A codebook was generated and used to index the text between two raters in order to improve the external credibility of the findings. Approximately 10% of the data was independently coded by two researchers (AB and CDL). The level of agreement between the two sets of analyses was measured using the Kappa coefficient (*k*), which was classified in line with the parameters proposed by Landis & Koch [15], including 0.8–1.0 = almost perfect; 0.6–0.8 = substantial; 0.4–0.6 = moderate; 0.2–0.4 = fair.

Sociodemographic information was analysed descriptively using SPSS® version 24. To test for differences between those included and excluded from the analysis the Chi-squared test *(χ2)* was used for categorical variables and the Mann-Whitney U test for continuous variables. All p-values presented are for two-tailed tests with $p < 0.05$ indicating statistical significance.

## Findings

### Demographics

The analytic sample included 653 individuals (16.5% of the total study sample). The mean age of those included in the analysis was 64.1 years (SD = 8.3), and 376 participants (57.6%) were aged ≤65 years. Compared to those exclude from the analysis, those included were significantly older ($U = 907221$, $p < 0.001$), but did not differ in gender distribution ($p = 0.31$).

Table 1 shows participants' sociodemographic data. The majority of participants were women (*n* = 459, 70.3%). Women were significantly younger than men *(U = 32503, $p < 0.001$)*. Most participants were Caucasian (*n* = 625, 95.7%; Male: 185, 28.3%; Female: 440, 67.4%). Many had a university degree (*n* = 394; 60.3%) and 272 (41.6%) were employed. Only 24 participants (3.7%) reported smoking, whereas 310 (47.5%) reported drinking alcohol most commonly 1 to 14 units per week.

### Qualitative results

From the qualitative analysis three overarching themes and eight subthemes were identified. The inter-rater reliability was high (*k* = 0.7). The themes included comments on motivation for lifestyle change, barriers to lifestyle change, and knowledge of dementia risk factors and behaviour change.

### Theme 1: Motivators to changing lifestyle

This theme included comments from 319 (48.8%) participants. Most (*n* = 179, 56.1%) were ≤ 65 years and 21.6% (*n* = 69) were women. Motivating factors were internally (e.g. feeling that dementia can be prevented through making life changes) or externally (e.g. family history) driven. Most respondents (*n* = 174, 54.5%) felt that having a family member with dementia could make them more susceptible to developing the condition, and 82 (25.7%) respondents felt that they would be more likely to engage in future lifestyle changes if they had a genetic predisposition to dementia (Sub-theme 1: Known family history):

> '*Because my Mum started with Alzheimer's aged 49, I do worry that I'm at high risk of getting. I would do anything to make any changes I feel would help.*' (Female, 53 years old)

**Table 1. Participants' sociodemographic data.**

| Demographics | Total Sample (n = 3948) (N, %) | Analytic Sub-sample (n = 653) (N, %) |
|---|---|---|
| **Age** (Mean, SD) | 62 (8.0) | 64.1 (8.3) |
| **Gender** (Female) | 2880 (72.9) | 459 (70.3) |
| **Ethnicity** (Caucasian) | 3835 (97.1) | 625 (95.7) |
| **Relationship status** | | |
| Married/Partnership | 2933 (74.2) | 459 (70.3) |
| Single/Separated/Divorced/Widowed | 1008 (25.5) | 192 (29.4) |
| Not reported | 7 (0.2) | 2 (0.3) |
| **Education** | | |
| Graduate and Postgraduate | 2297 (58.2) | 394 (60.3) |
| Non-graduate | 1651 (41.8) | 178 (27.2) |
| **Work status** (Yes) | 1958 (49.6) | 272 (41.6) |
| **Ever known someone living with dementia** (Yes) | 3085 (78.1) | 607 (93.0) |
| **Ever been a carer for someone with dementia?** (Yes) | 1776 (45.0) | 368 (56.3) |
| **Smoker** (Yes) | 126 (3.2) | 24 (3.7) |
| **Alcohol use** (Weekly units) | | |
| Low risk (1 to 14 units) | 2732 (69.2) | 310 (47.5) |
| At risk (>14 units) | 795 (20.1) | 274 (42.0) |

*'Having watched my mum suffer got [sic] over 4 years I think this was the wake-up call to change attitudes to living.'* (Male, 62 years old)

Willingness to change lifestyle was related to a positive perception around the benefits of such changes and being in control of their own life (Sub-theme 2: Perceived benefits of a 'healthy' lifestyle). For example, 71 (22.2%) respondents reported they would be willing to take new advice about dementia on board, even though they felt that they were already following a 'healthy' regime:

*'I have taken most of the steps usually indicated. I am quite willing to make further changes if they are identified.'* (Female, 67 years old)

*'I feel that I have a healthy lifestyle (perhaps a little too much alcohol and sweet food). If I was told by an expert that, in fact, my lifestyle needs to change quite considerably then I would do so.'* (Male, 57 years old)

## Theme 2: Barriers for lifestyle change

This theme included comments from 168 (25.7%) participants of whom 102 (60.7%) were aged ≤ 65 years and 98 (58.3%) were women. Dementia was mostly perceived by participants as a threat but 59 (35.1%) reported that they gave their health priority to other conditions such as diabetes, neuropathy and chronic pain and these were barriers for behaviour change in relation to dementia (Sub-theme 3: Competing health priorities):

*'As a diabetic since I was 23 makes it difficult to change my lifestyle. I do my best living with diabetes and neuropathy. Exercise is almost impossible as I have great difficulty in walking.'* (Female, 70 years old)

*'I do not think about dementia; my physical health concerns me. Back pain, a bad knee, forced me to reduce vigorous activity (such as football). I fear having cancer much more than anything else.'* (Male, 63 years old)

Some participants (*n* = 63, 37.5%) reported that caring for someone took away much of their strength, motivation and free time to engage in any healthy behaviours, such as physical activity or social engagement (Sub-theme 4: Time constraints due to caring responsibilities):

*'No idea whether I am likely to develop dementia. Some answers—bad sleep, stress, fatigue—are because I am a full-time carer for a husband with a progressive physical condition.'* (Female, 67 years old)

*'I could do a little more strenuous activity but busy caring for elderly parent and young grandchildren as well as working leaves little time.'* (Female, 61 years old)

Several participants (n = 79, 47.0%) reported scepticism about focusing on risk factors for dementia prevention because they felt there was a lack of good-quality evidence around specific risk factors and their respective guidelines (e.g. number of alcohol units per day, amount of sugar intake). Respondents felt that evidence was mostly inconclusive and did not differ from risk factors for any other health conditions (e.g. diabetes and heart disease). For example, some participants felt that controlling their sugar intake could decrease their risk of developing diabetes and also dementia later in life (Sub-theme 5: Scepticism around risk factors):

*'I found it difficult to be very positive about changing my habits to avoid dementia because the advice on which habits are bad for you is not very conclusive.'* (Female, 66 years old)

*'Most of my responses would be the same if related to other medical conditions e.g. Reducing chances of a heart attack or reducing chances of physical deterioration—joints etc.'* (Male, 63 years old)

Some respondents (*n* = 16, 9.5%) felt that because they were already following a 'healthy' regime, they were unlikely to make any further improvements in lifestyle for dementia risk reduction (Sub-theme 6: Already living a healthy lifestyle therefore reluctant to do more):

*'I feel that I have a pretty healthy lifestyle already and it is doubtful that I will devote more effort to make this healthier.'* (Female, 57 years old)

*'I do meditation, I exercise and have a good diet, low in carbohydrate and with enough omega 3. . .so I am not so confident I can make changes.'* (Male, 60 years old)

### Theme 3: Quality of the information received

This theme included comments from 179 participants (27.4%); 104 (58.1%) were aged ≤ 65 years (Mean = 57.7; SD = 4.3) and 78 (43.6%) were women. The theme refers to the quality of and trust in the information about risk factors received from different sources such as health professionals, health-related and mainstream TV programs, and weekly magazines. Receiving evidenced-based information on the risk factors for dementia was reported by some respondents (*n* = 57, 31.8%) as being key to changing lifestyle to reduce dementia risk (Sub-theme 7: Evidence-based information):

'. . .I've studied the topic extensively, in academic papers, books, videos, on-line, etc. I've made massive changes to my lifestyle over the last 2 or 3 years, many of them suitable for dementia-risk-reduction. For example, I use a Ketogenic Diet in order to run my brain mainly on ketones.' (Male, 70 years old)

'Depends on whether there was proven research into what is being suggested. Maybe GPs should do a dementia awareness appointment for all over 60 to give individual advice.' (Female, 63 years old)

On the other hand, a similar number of participants (*n* = 63, 35.2%) felt that the messages received were not trustworthy, especially in reference to the risk of developing dementia and the need for preventative measures. Hence, people were sceptical about making changes to their lifestyle (Sub-theme 8: Low levels of trust in the information coming from the media):

'I found it difficult to be very positive about changing my habits to avoid dementia because the advice on which habits are bad for you is not very conclusive.' (Female, 66 years old)

'There is a bewildering amount of information in the media about dementia and its causes. It's difficult to decipher what is true or what really needs to be taken seriously.' (Male, 55 years old)

## Discussion

This large-scale qualitative analysis helps better understand the underlying views about behaviour change and dementia risk. Three main themes and eight subthemes were identified. These reflected barriers as well as facilitators to making lifestyle changes for dementia risk reduction. Having a known family history of dementia, having a positive perception of the benefits of a healthy lifestyle and trusting the information on dementia provided by different sources appeared to motivate people to change behaviour. In contrast, time constraints due to caring responsibilities, having other existing or foreseen health conditions, being sceptical about risk factors for dementia, receiving poor quality information on dementia, and feeling that a healthy regime is already in place, reduced willingness to change. Consequently, having a 'healthy' lifestyle may act as either an enhancer for future behaviours to reduce dementia (e.g. being willing to engage in more physical activity), or as de-motivator for engaging in further preventative measures.

Our results partly fit with the Health Belief Model for behaviour change [16]. This posits that in the presence of a healthier lifestyle, the person may perceive making small adjustments as easier to commit to, whereas individuals who engage in numerous health risk behaviours (e.g. obesity, smoking, excess alcohol intake) may perceive greater barriers to change.

Our previous survey results looking at willingness to reduce alcohol intake for dementia risk reduction (*reference omitted for blinded peer review*) showed a positive relationship between having a healthy lifestyle and willingness to reduce alcohol intake (e.g. engaging in physical activity). However, some people who believed they were already leading a healthy lifestyle reported feeling less inclined to change their lifestyle further. It could be that these individuals are in fact already leading a low risk lifestyle and they should be encouraged by health providers to continue to do so. However, misperceptions between current and expected physical activity levels and diet quality have been found in previous population studies, particularly in older people with lower education [17,18,19]. Therefore, health professionals should work closely with these individuals so that they have an accurate view of their current health behaviour and can implement improvements when necessary.

The study identified a large number of barriers to change. Although there is potentially a linear trend for behaviour change (the healthier the person reports their lifestyle to be, the easier it may be for them to make other changes) a small number of participants ($n = 16$) self-reported a saturation point for health over which no more adjustments were thought to be required. However, it is not clear "how much is enough" to maximise dementia risk reduction, especially owing to the lack of evidence-based guidance with respect to specific parameters for health behaviour change (e.g. number of miles to walk per day, number of alcohol units per week). One explanation could be what Kessler et al [12] defined as 'dementia worry', as akin to the fight-flight response to danger. That is, the person may be willing to fight (or change lifestyle) if the fear or the energy required to deal with the stimulus (the danger) is not too high. Conversely, when the anxiety produced by the feared stimulus (engage in dementia risk reduction) is too high or it requires a high level of energy to be consumed, the person may be more willing to opt for a flight response (denial or avoidance of behaviour change). Another explanation could be linked to the socioemotional selectivity theory for health behaviour change [20]. According to this theory there are considerable age-related changes across life when people make a decision to change their own lifestyle. As people age they tend to favour present-oriented emotional goals over future-oriented goals. Hence, the older the person the less likely they will be to make changes to reduce risk of dementia. In the context of multiple health priorities and with exposure to various messages regarding disease prevention, greater clarity is needed on the factors as well as decision-making processes that drive people over 50 towards making additional lifestyle changes.

We also found that not considering dementia to be a priority was a barrier to lifestyle change. However, it can be very difficult to encourage people to make lifestyle changes for a disease that may be decades away and especially when the causes of dementia are not yet fully understood. A recent national survey conducted in the USA on individual perceptions of health conditions [21], in a sample of 4,033 respondents, found that cancer ranked as the most feared condition (40.3%) compared to dementia (17.5%). However, recent changes in the leading cause of death stating that dementia has overtaken cancer and cardiovascular disease, may change these perceptions in the future [22, 23].

Disease co-morbidity was also a barrier to lifestyle change. We identified cases in which participants with cancer would not engage in preventative measures for dementia because the former was given priority over the less immediate risk of the latter. Such hierarchical priorities among health conditions may be explained in terms of the 'here-and-now' hypothesis for individual proneness to behaviour change [24]. It could be argued that the preoccupation of coping with an existing health condition is directly associated with the extent to which the person takes precautionary measures to prevent the onset of dementia. In other words, the more the preoccupation towards an existing condition (e.g. cancer or diabetes) the less the motivation to adequately engage in behaviour risk management for future health problems (e.g. dementia). It should be noted that some risk factors may be associated with multiple health conditions. Hence, there may be cases in which a person tackling one risk factor for cancer, may already be working proactively towards preventing dementia, and vice versa. It may also be that people's belief that dementia only affects individuals very late in life would reduce the need for immediate action compared to cancer, which is more known to affect people at any age.

Last, we found information source and its integrity could potentially impact behaviour change. The results support the here-and-now hypothesis for prevention, whereby the less information a person with an existing condition is provided about the risks of developing dementia, the less the person will be motivated to address them, by focusing more on the here-and-now (existing) condition. It is to be acknowledged that public health campaigns about

dementia mostly target people aged 60 and over [25], which may reinforce the false idea that dementia prevention needs to occur only in later life. Older people are also more likely to have comorbid diagnoses and find it difficult to properly address risk factors specifically targeting dementia (e.g. increase physical activity in the presence of physical disability). This may also impact their motivation to replace existing maladaptive behaviours for example, as the person may feel overwhelmed with dealing with other health conditions.

The negative impact that misleading information regarding dementia portrayed by the media may have on people's adherence to healthier lifestyles for dementia risk reduction was strongly identified in our survey (Sub-theme 8). This corroborates findings from previous analyses of the UK media representations of dementia [23, 26] which found that negative messages around dementia may lead to stigmatisation and misinformation and to the promotion of a dichotomy representing dementia in terms of "killer" and "victim". Our work extends the study by Peel [27] who defined such messages as "panic-blame", which may lead to individual culpability, similar to what we found in our survey. Future research is needed to conduct a media analysis of how the representation of dementia is shifted over time, which could help identify those social media discourses that influenced for example the spread of information around prevention, made in terms of culpability. A media analysis could also help establish the presence of real-life stories of people with dementia backing up the reporting on dementia prevention and behaviour change, as little active involvement of people with dementia is promoted in media coverage.

Many participants felt that the media was not trustworthy. This corroborates previous literature on public opinion on health information [28] in which older people reported trusting sources of health information in the following order (from higher to lower levels of trust): health care providers, pharmacists, friends and relatives, retirement community staff, newspapers, the internet, television, and the radio [27]. We argue that public health campaigns should strive for evidence-based information on dementia in the media and provide support to better access trusted online sources.

## Strengths and limitations

The strengths of this study include the large sample of respondents providing free-text comments for analysis and the richness of information regarding participants' perceptions around possible motivators and barriers to make changes to lifestyle. The use of such boxes in the survey acted as a catchment to collect participants' views without the restrictions imposed by standardised questionnaires which could have influenced participants' responses through the use of biased prompts.

There are some limitations, and even though this study has helped us understand attitudes to lifestyle and dementia risk, more in-depth work is needed to contribute to better public health approaches. While participants weren't specifically educated about risk reduction and behaviour change, before taking part in the study, they completed a survey that did ask specific questions about some lifestyle aspects, and this may have influenced their responses.

The sample is self-selected, highly educated and the majority are Caucasian. Therefore, caution is needed in interpreting the results as they may not reflect the views and experiences of those living in deprived areas or individuals from other ethnic groups. Only 20.0% of all survey respondents provided their views on behaviour change for dementia risk reduction. This could limit the generalisability of our findings as we only captured the views of those highly motivated to share them. It is interesting to see that other possible barriers to change such as cost (e.g. for a healthier diet, gym membership/exercise equipment) or a sedentary job were not reflected in the findings and some respondents may have been reluctant to disclose their

views. In addition, the character limits in the text-box may have affected the amount of information that respondents could include. Also, respondents may have not recalled some particular ideas around behaviour change and risk reduction at the time they were asked to answer the survey.

Furthermore, our interpretation is time-bound and individuals' perceptions and beliefs around behaviour change and dementia risk reduction may change over time and through experiences. The use of free text boxes could have limited the validity of our findings from the content analysis, as no standardised line of inquiry was performed, and no further prompts were made to elicit a richer narrative from respondents. A follow-up study may wish to explore these themes using semi-structured interviews to gather additional detail from a more diverse group of people.

## Implications for dementia prevention

While ageing is a strong predictor of cognitive decline, dementia is not a consequence of ageing, but rather is associated with non-modifiable (e.g. genetic makeup, gender, ethnicity) and modifiable (e.g. alcohol use, obesity, hypertension) risk factors. If modifiable risk factors are properly addressed through public health prevention programs, risk of future dementia could be decreased [29, 30]. The global action plan on the public health response to dementia 2017–2025 [1] lists seven priority areas to target the impact of dementia on people with dementia, their families and at the community level, and to reduce the risk to develop the condition over time. These areas are: dementia as a public health priority, awareness and friendliness, risk reduction, diagnosis treatment and care, support for carers, information system for dementia, research and innovation. Our findings point to the importance of delivering evidence-based information at the population levels through easily accessible media programs (i.e. TV health programs, social media adverts). The need for these health programs is aligned with the risk reduction guidelines and WHO mandate to effectively support research on dementia risk reduction and improve the capacity of health and social care personnel to provide specific population-based interventions around modifiable factors (e.g. increase physical exercise) that are associated with other non-communicable diseases. This information would reduce the confusion around the overlap that some modifiable factors may have with multiple conditions. Evidence-based information may further educate individuals on healthy regimes that could help prevent dementia, for example by reporting on how many alcohol units may be safe to have and/or which type of exercises are most effective to maintain a healthy brain [1]. Population-based campaigns would need to be accompanied by programs that take into account the variabilities of cases (e.g. pre-existing and or comorbid medical conditions) and time restrains (e.g. due to caring responsibilities).

## Conclusions

The study identified what helps people engage in preventative measures to reduce the risk of dementia later in life including having a personal experience of dementia and perceiving oneself as having a 'healthy' lifestyle. There may be unique challenges in later life that interact with the likelihood of behaviour change such as living with long-term conditions and having caring responsibilities. The quality of information received around dementia prevention acted as a motivating factor when it was backed up by evidence-based data, and as a barrier to change when the sources were not trusted (e.g. TV, radio, newspapers). The latter was found to spread negative representations of dementia and diffuse a sense of culpability in the person attempting to change their own lifestyle. Considering the highly individual variability in engaging in behaviour change for dementia risk reduction, and the high variability of information in the

media around dementia, this means that prevention programs should help people better access trusted sources of information to acquire knowledge on risks for dementia and what preventative measures can be taken to avoid developing the condition.

## Author Contributions

**Conceptualization:** Alessandro Bosco, Katy A. Jones, Martin Orrell, Deborah Oliveira.

**Data curation:** Deborah Oliveira.

**Formal analysis:** Alessandro Bosco, Claudio Di Lorito.

**Methodology:** Alessandro Bosco, Katy A. Jones, Claudio Di Lorito, Blossom C. M. Stephan, Martin Orrell, Deborah Oliveira.

**Resources:** Deborah Oliveira.

**Supervision:** Blossom C. M. Stephan, Martin Orrell.

**Validation:** Alessandro Bosco, Claudio Di Lorito.

**Writing – original draft:** Alessandro Bosco.

**Writing – review & editing:** Alessandro Bosco, Katy A. Jones, Claudio Di Lorito, Blossom C. M. Stephan, Martin Orrell, Deborah Oliveira.

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
