## [Decision Letter · Decision Letter 0]

24 Mar 2020

PONE-D-20-04265

Changing lifestyle for dementia risk reduction: Inductive content analysis of a national UK survey

PLOS ONE

Dear Dr Bosco,

Thank you for submitting your manuscript to PLOS ONE. After careful consideration, we feel that it has merit but does not fully meet PLOS ONE’s publication criteria as it currently stands. Therefore, we invite you to submit a revised version of the manuscript that addresses the points raised during the review process.

We would appreciate receiving your revised manuscript by April 30, 2020. To enhance the reproducibility of your results, we recommend that if applicable you deposit your laboratory protocols in protocols.io, where a protocol can be assigned its own identifier (DOI) such that it can be cited independently in the future. For instructions see: http://journals.plos.org/plosone/s/submission-guidelines#loc-laboratory-protocols

We look forward to receiving your revised manuscript.

Kind regards,

Mellissa H Withers, PhD, MHS

Academic Editor

PLOS ONE

Journal Requirements:

2.  We noted in your submission details that a portion of your manuscript may have been presented or published elsewhere. 

[No. However this is a secondary data analysis and to give context to the analysis, we briefly report on data from the primary analysis of a large-scale survey that we published in: Oliveira D, Jones KA, Ogollah R, Ozupek S, Hogervorst E, Orrell M. Willingness to adhere to current UK low-risk alcohol guidelines to potentially reduce dementia risk: A national survey of people aged 50 and over. Journal of Alzheimer's Disease. 2019 Jan 1;69(3):829-37.]

i) Please clarify whether this [ publication] was peer-reviewed and formally published. If this work was previously peer-reviewed and published, in the cover letter please provide the reason that this work does not constitute dual publication and should be included in the current manuscript.

Reviewers' comments:

Reviewer's Responses to Questions

**Comments to the Author**

1. Is the manuscript technically sound, and do the data support the conclusions?

Reviewer #1: Yes

Reviewer #2: Yes

2. Has the statistical analysis been performed appropriately and rigorously? 

Reviewer #1: Yes

Reviewer #2: Yes

3. Have the authors made all data underlying the findings in their manuscript fully available?

Reviewer #1: Yes

Reviewer #2: No

4. Is the manuscript presented in an intelligible fashion and written in standard English?

Reviewer #1: Yes

Reviewer #2: Yes

5. Review Comments to the Author

Reviewer #1: This study contributes to a growing body of knowledge about risk reduction behavior with a focus on dementia. It uses an innovative approach in analyzing comments submitted anonymously by persons who completed an on line survey. The use of on line research provides the potential benefit of being able to expand participation in research to a wider audience. However, in this case, the respondents were overwhelmingly Caucasian, well educated and women which limits the generalizability of the findings (as noted by the authors).

A majority of the population of respondents were under age 65 (56%/61%/58% between 50-65) for each of the three themes identified. This represents a late middle age or early age population where interventions for risk reduction for late life dementia have the potential to be most effective. The authors point out (pp12-12) that “public health campaigns about dementia mostly target people aged 60 and over, which may reinforce the false idea…”Indeed, lifestyle change in later life may occur too late to be effective in delaying onset or reducing risk for dementia. The need to rethink public health campaigns based on current knowledge would suggest targeting adults, young and middle age, to make lifestyle changes that reduce risk have great potential. However, as is well known in other chronic diseases (hypertension, diabetes, etc) it can be extremely difficult to get individuals to make such changes for a potential disease that is many years, and often decades, away.

A few specific items:

Pg 4, line 84 – the word ‘improve’ carries with it a value statement; I’d suggest ‘change’ would be a better choice

Pg 6, line 124 – sentence is incomplete, needs a final word, should read “…. specific group they belonged to.”

Pg 13, line 321 – the word “thrive” is incorrect, should be ‘strive’

Pg 13, line 322 and pg 15, line 373– why only “older people” – as noted above, it may be more beneficial to target campaigns at young and middle aged adults.

Reviewer #2: This paper aims to investigate the motives, barriers, and information sources for a lifestyle change to reduce the risk of dementia. This is a crucial step to acquire this information if aiming to launch a lifestyle changing campaign. I have several questions for the authors to clarify.

1. Although the current results were understandable as motives, barriers, and trustable information source from evidence-based research, I am surprised that other possible barriers were not reflected, such as financial reason (people may need to pay more for a healthy diet, the cost in the fitness center or exercise equipment/court, etc.), a sedentary job that has to put much time, unawareness their health condition or unawareness what are bad for cognition. Could the bias be generated by the possibility that responders unreluctantly tell specific reasons, character limits in the text-box, not being able to call to mind particular ideas at the time answering the survey, etc.?

2. Line 193-196, The context 'Several participants (n=79, 47.0%) reported skepticism about focussing on risk factors specific to dementia as they felt that these did not differ from risk factors for any other health conditions. For example, some participants felt that controlling their sugar intake could decrease their risk of developing diabetes and also dementia later in life (Sub-theme 5: Scepticism around risk factors)'.

The reality is these factors are indeed not only specific to dementia but also risk factors for many other health conditions such as CHD, CVD, peripheral artery disease. I don't quite understand why this thought is a barrier to a lifestyle change. Given they are risk factors for many other conditions, in common sense, it should be one of the motives to control these factors.

3. During the survey, whether the participants have been instructed that these factors have been proposed as risk factors for dementia? This is the precondition for people to consider a lifestyle change. As stated in the manuscript, 'In the UK, a national survey of 2,361 adults aged ≥15, found that only 34% believed the risk of dementia could be reduced through lifestyle changes, and 22% thought dementia was an inevitable consequence of ageing', The conditions 'don't know' or 'know but don't believe' could affect differently on people's choice of lifestyle change.

In general, the idea/topic is worthy of being investigated. Still, I am a little concerned about the capability of this study to reflect this issue comprehensively before a lifestyle change campaign. More detailed designs specific to this topic may need.

6. PLOS authors have the option to publish the peer review history of their article (what does this mean?). If published, this will include your full peer review and any attached files.

Reviewer #1: No

Reviewer #2: No

---

## [Author Response · Author response to Decision Letter 0]

3 Apr 2020

1) Did the authors present any new data in this submission that were not previously presented in the published article: Oliveira D, Jones KA, Ogollah R, Ozupek S, Hogervorst E, Orrell M. Willingness to adhere to current UK low-risk alcohol guidelines to potentially reduce dementia risk: A national survey of people aged 50 and over. Journal of Alzheimer's Disease. 2019 Jan 1;69(3):829-37.'?

1) Response: We confirm that this manuscript does not constitute dual publication. It contains data that were not previously published and based on a new analysis of free text boxes of a large survey study. As our analysis utilises a subsample of a larger sample, we only briefly report on the number of the total sample and its key socio-demographics to contextualise our secondary analysis for the sub-analytic sample. The primary analysis of a large-scale survey was previously published by the authors in: Oliveira D, Jones KA, Ogollah R, Ozupek S, Hogervorst E, Orrell M. Willingness to adhere to current UK low-risk alcohol guidelines to potentially reduce dementia risk: A national survey of people aged 50 and over. Journal of Alzheimer's Disease. 2019 Jan 1;69(3):829-37. Please see our response to the editor's comment in the cover letter. 

2) Did the authors perform any additional experiments or collect any additional data that were not a part of the study from the published article: Oliveira D, Jones KA, Ogollah R, Ozupek S, Hogervorst E, Orrell M. Willingness to adhere to current UK low-risk alcohol guidelines to potentially reduce dementia risk: A national survey of people aged 50 and over. Journal of Alzheimer's Disease. 2019 Jan 1;69(3):829-37.'?

2) Response: We run a secondary analysis of data from participants who took part in a large survey study, however, the analysis we run was novel and was not published elsewhere. We did not recruit new participants for the analysis and we used a sub analytic sample, as this was a secondary analysis, but the findings are novel as we undertook analysis on data that were not previously analysed. All our findings represent new material and novel contribution to knowledge. In the only occasion where we mentioned data linked to the larger sample we referenced the publication so we confirm there is no risk to incur in dual publication. 

Thank you

The authors

---

## [Decision Letter · Decision Letter 1]

28 Apr 2020

Changing lifestyle for dementia risk reduction: Inductive content analysis of a national UK survey

PONE-D-20-04265R1

Dear Dr. Bosco,

We are pleased to inform you that your manuscript has been judged scientifically suitable for publication and will be formally accepted for publication once it complies with all outstanding technical requirements.

With kind regards,

Mellissa H Withers, PhD, MHS

Academic Editor

PLOS ONE

Additional Editor Comments (optional):

Reviewers' comments:

Reviewer's Responses to Questions

**Comments to the Author**

1. If the authors have adequately addressed your comments raised in a previous round of review and you feel that this manuscript is now acceptable for publication, you may indicate that here to bypass the “Comments to the Author” section, enter your conflict of interest statement in the “Confidential to Editor” section, and submit your "Accept" recommendation.

Reviewer #1: All comments have been addressed

Reviewer #2: All comments have been addressed

2. Is the manuscript technically sound, and do the data support the conclusions?

Reviewer #1: Yes

Reviewer #2: Yes

3. Has the statistical analysis been performed appropriately and rigorously? 

Reviewer #1: Yes

Reviewer #2: Yes

4. Have the authors made all data underlying the findings in their manuscript fully available?

Reviewer #1: Yes

Reviewer #2: No

5. Is the manuscript presented in an intelligible fashion and written in standard English?

Reviewer #1: Yes

Reviewer #2: Yes

6. Review Comments to the Author

Reviewer #1: Thank you for addressing all comments and making suggested language/word changes in the revised manuscript.

Reviewer #2: I did not find the response from the authors to my comments specifically. But I read the relevant changes to my comments in the manuscript. I do not have further comments to the authors.

7. PLOS authors have the option to publish the peer review history of their article (what does this mean?). If published, this will include your full peer review and any attached files.

Reviewer #1: No

Reviewer #2: No

---

## [Editor Report · Acceptance letter]

5 May 2020

PONE-D-20-04265R1 

Changing lifestyle for dementia risk reduction: Inductive content analysis of a national UK survey 

Dear Dr. Bosco:

I am pleased to inform you that your manuscript has been deemed suitable for publication in PLOS ONE. Congratulations! Your manuscript is now with our production department. 

With kind regards,

on behalf of

Dr. Mellissa H Withers 

Academic Editor

PLOS ONE